# Functional and antigenic characterization of SARS-CoV-2 spike fusion peptide by deep mutational scanning

Ruipeng Lei [1,13], Enya Qing [2,13], Abby Odle[3], Meng Yuan [4], Chaminda D. Gunawardene[5,6], Timothy J. C. Tan [7], Natalie So[1,8], Wenhao O. Ouyang[1], Ian A. Wilson [4,9], Tom Gallagher [2] ✉, Stanley Perlman [3,10] ✉, Nicholas C. Wu [1,7,11,12] ✉ & Lok-Yin Roy Wong [3,5,6] ✉

The fusion peptide of SARS-CoV-2 spike protein is functionally important for membrane fusion during virus entry and is part of a broadly neutralizing epitope. However, sequence determinants at the fusion peptide and its adjacent regions for pathogenicity and antigenicity remain elusive. In this study, we perform a series of deep mutational scanning (DMS) experiments on an S2 region spanning the fusion peptide of authentic SARS-CoV-2 in different cell lines and in the presence of broadly neutralizing antibodies. We identify mutations at residue 813 of the spike protein that reduced TMPRSS2-mediated entry with decreased virulence. In addition, we show that an F823Y mutation, present in bat betacoronavirus HKU9 spike protein, confers resistance to broadly neutralizing antibodies. Our findings provide mechanistic insights into SARS-CoV-2 pathogenicity and also highlight a potential challenge in developing broadly protective S2-based coronavirus vaccines.

While the world is slowly returning to normal from the COVID-19 pandemic, severe acute respiratory syndrome coronavirus 2 (SARS-CoV-2) continues to circulate in the human population. Due to the importance of COVID-19 vaccine development, spike (S) is the most studied SARS-CoV-2 protein. S facilitates virus entry by binding to the host receptor angiotensin-converting enzyme 2 (ACE2) and mediates virus-host membrane fusion by undergoing drastic conformational changes[1]. Membrane fusion is activated by the cleavage of the S2′ site in the S2 domain by either TMPRSS2 at the cell surface or cathepsins in the endosome[2–4]. With cleavage of the S2′ site, the fusion peptide is exposed and inserted into the membrane of the host cell[5]. Subsequently, the S2 domain rearranges into a stable six-helix bundle with a long central three-stranded coiled coil to complete membrane fusion[6,7]. Although early SARS-CoV-2 variants enter cells mainly by TMPRSS2-mediated cleavage, some Omicron variants have been shown to utilize cathepsin-mediated endosomal entry[8–11]. This shift of cell entry pathway may associate with changes in cellular tropism and reduction in virulence[8,9]. As a result, studying the determinants of SARS-CoV-2 membrane fusion has important public health implications.

[1]Department of Biochemistry, University of Illinois at Urbana-Champaign, Urbana, IL 61801, USA. [2]Department of Microbiology and Immunology, Loyola University Chicago, Maywood, IL 60153, USA. [3]Department of Microbiology and Immunology, University of Iowa, Iowa City, IA 52242, USA. [4]Department of Integrative Structural and Computational Biology, The Scripps Research Institute, La Jolla, CA 92037, USA. [5]Center for Virus-Host Innate Immunity, Rutgers New Jersey Medical School, Newark, NJ 07103, USA. [6]Department of Microbiology, Biochemistry and Molecular Genetics, Rutgers New Jersey Medical School, Newark, NJ 07103, USA. [7]Center for Biophysics and Quantitative Biology, University of Illinois at Urbana-Champaign, Urbana, IL 61801, USA. [8]Department of Computer Science, University of Illinois at Urbana-Champaign, Urbana, IL 61801, USA. [9]The Skaggs Institute for Chemical Biology, The Scripps Research Institute, La Jolla, CA 92037, USA. [10]Department of Pediatrics, University of Iowa, Iowa City, IA 52242, USA. [11]Carl R. Woese Institute for Genomic Biology, University of Illinois at Urbana-Champaign, Urbana, IL 61801, USA. [12]Carle Illinois College of Medicine, University of Illinois at Urbana-Champaign, Urbana, IL 61801, USA. [13]These authors contributed equally: Ruipeng Lei, Enya Qing. ✉e-mail: tgallag@luc.edu; stanley-perlman@uiowa.edu; nicwu@illinois.edu; roy.wong@rutgers.edu

Residues 816–834 of the S protein, which is located immediately downstream of the S2′ cleavage site at Arg815/Ser816[12], have generally been recognized as the bona fide SARS-CoV-2 fusion peptide (bFP, residues 816–834)[13–15]. Nevertheless, a recent cryo-EM structure of the postfusion SARS-CoV-2 S in a lipid bilayer membrane showed that the internal fusion peptide (iFP, residues 867–909) inserted into the membrane, whereas the bFP was not resolved[16]. This observation appears to challenge the functional importance of bFP but also indicates that additional analysis of the fusion peptide and fusion mechanism of SARS-CoV-2 S is warranted.

Neutralizing antibodies targeting the functionally important S2 domain have been isolated from convalescent individuals[17–21]. Unlike antibodies to the immunodominant receptor-binding domain (RBD) of S1[22,23], S2 antibodies typically have very broad cross-reactivity due to high S2 sequence conservation[17–21]. Neutralizing antibodies to an epitope that spans the S2′ cleavage site and the bFP can cross-react with diverse coronavirus strains from all four genera (α, β, γ, and δ)[17,20,21,24]. These broadly neutralizing antibodies provide important insights into the development of a pan-coronavirus vaccine. However, comprehensive assessments of the genetic barrier for resistance to bFP antibodies have not been completed. Relatedly, the mutational tolerance of the SARS-CoV-2 bFP is largely elusive.

Deep mutational scanning, which combines saturation mutagenesis and next-generation sequencing, allows the phenotypes of many mutations to be measured in parallel. Deep mutational scanning has been applied to study the mutational fitness effects of various medically important RNA viruses, including influenza virus[25,26], human immunodeficiency virus[27], hepatitis C virus[28], and Zika virus[29]. All of these viruses can be evaluated using efficient plasmid-based reverse genetic systems, which are prerequisites for applying deep mutational scanning to study viral replication fitness. At the same time, most, if not all, deep mutational scanning studies of SARS-CoV-2 have been performed using protein display or pseudovirus systems[30–33]. Although these studies have offered critical insights into antibody resistance and biophysical constraints of SARS-CoV-2 evolution, they do not directly measure virus replication fitness or virulence. While multiple reverse genetic systems are available for SARS-CoV-2[34–36], they are more complex than those for other RNA viruses, mainly due to the larger genome size of SARS-CoV-2. Thus, probing the fitness effects of SARS-CoV-2 mutations by deep mutational scanning can be technically challenging.

In this study, we perform deep mutational scanning of S residues 808–855, spanning the S2′ cleavage site, bFP, and fusion peptide proximal region (FPPR)[16], using a bacterial artificial chromosome (BAC)-based reverse genetic system of SARS-CoV-2. Our results reveal that the bFP (residues 816–834) has a very low mutational tolerance. In addition, we identify mutations upstream of the S2′ cleavage site that reduced TMPRSS2-mediated entry. Further characterizations of these mutations suggest a relationship between sensitivity for TMPRSS2-mediated S2′ cleavage, cell entry pathway, and virus virulence. We also identify a mutation in the bFP that resists two broadly neutralizing bFP antibodies and naturally exists in a bat coronavirus strain.

## Results

### Deep mutational scanning of SARS-CoV-2 bFP

Based on a BAC-based reverse genetic system of SARS-CoV-2 Wuhan-Hu-1 (pBAC SARS-CoV-2)[37,38], we constructed a saturation mutagenesis library that contained all possible single amino acid mutations in the bFP and FPPR (residues 816–855) of the SARS-CoV-2 S, as well as the eight residues immediately upstream of the S2′ cleavage site (residues 808–815)[38]. The BAC mutant library was transfected into Vero cells to generate a virus mutant library, which was then passaged once in Calu-3 or Vero cells for 48 h. The frequencies of individual mutations in the BAC mutant library and the post-passaged mutant library were determined by next-generation sequencing (Fig. S1). The fitness value of

each mutation was calculated based on its frequency enrichment and normalized such that the mean fitness values of silent mutations and nonsense mutations were 1 and 0, respectively (see Methods). The fitness values of 893 (98%) out of 912 all possible amino acid mutations across the 48 residues of interest were measured (Fig. 1). Pearson correlation coefficients of 0.62 (Calu-3) and 0.58 (Vero) were obtained between two biological replicates (Fig. S2A, B), demonstrating the reproducibility of our deep mutational scanning experiments. Moreover, the fitness value distributions of silent and nonsense mutations had minimal overlap, further validating our results.

To further characterize our BAC-based reverse genetic system for deep mutational scanning, we also analyzed the post-transfection mutant library by next-generation sequencing. The result indicated that fitness selection was present at the transfection step as the fitness values of silent mutations were significantly higher than nonsense mutations in the post-transfection mutant library ($p < 0.0001$, Fig. S3A). Besides, the fitness values of the post-transfection mutant library had a Pearson correlation coefficient of 0.68 between replicates (Fig. S3B), suggesting that a genetic bottleneck took place at the step of transfection, albeit mild, in our deep mutational scanning experiments. Of note, our mutant library construction approach was designed to prevent secondary mutations (see Methods). Consistently, >90% of the sequencing reads of the BAC mutant library had either 0 or 1 mutation (Fig. S3C). To assess the mutation rate of the BAC mutant library without confounding by sequencing errors from next-generation sequencing, we sequenced 22 individual clones of the BAC mutant library. Among the 22 clones, 20 (91%) had 0 or 1 mutation, two clones (9%) had 2 mutations, and none had >2 mutations (Fig. S3D). This result substantiates that our mutagenesis approach yielded predominantly 1 mutation per clone.

Recently, the effects of ~7000 natural mutations of SARS-CoV-2 S on cell entry have been quantified by a pseudovirus-based deep mutational scanning experiment[33]. The fitness effects of natural mutations in the SARS-CoV-2 genome have also been estimated using a phylogenetic-based approach in another study[39]. Although these studies only examined <50% of all possible amino acid mutations from residues 808 to 855, their measurements moderately correlated with our deep mutational scanning results (rank correlation ranges from 0.36 to 0.49, Fig. S2G–J). Of note, while natural mutations in circulating SARS-CoV-2 have been observed at each of residues 808–855, their natural occurrence frequency is all less than 0.3%[40].

### Mutations at residue 813 modulate protease utilization for S2′ cleavage

Based on our deep mutational scanning results, we observed that certain mutations had high fitness values in Vero cells but not in Calu-3 cells (Fig. 1). This observation was particularly apparent at residue 813, which is upstream of the S2′ cleavage site. Coronaviruses, including SARS-CoV-2, are known to enter Calu-3 cells through TMPRSS2-mediated membrane fusion on the cell surface[10,41,42]. In contrast, coronaviruses enter Vero cells, with low TMPRSS2 expression, through cathepsin-mediated membrane fusion in endosomes[10,42–44]. As a result, we hypothesized that mutations at residue 813 shifted the preference of protease utilization for the S2′ cleavage site.

To test this hypothesis, we generated VSV-based pseudoparticles (VSVpps) bearing wild-type (WT), S813V, or S813K SARS-CoV-2 S. Although S813V and S813K slightly decreased the incorporation of S into VSVpp (Fig. S4A), their efficiency of Vero cell entry was similar to WT when entry mostly occur through endosomal entry (Fig. 2A). Furthermore, cathepsin inhibitor E64D, but not TMPRSS2 inhibitor camostat, significantly reduced Vero cell entry to a greater extent in S813V and S813K compared to WT (Fig. 2C, D), showing that S813V and S813K were more sensitive to blockade of endosomal entry. When Vero cells overexpressed TMPRSS2 at the cell surface (Vero-TMPRSS2), both

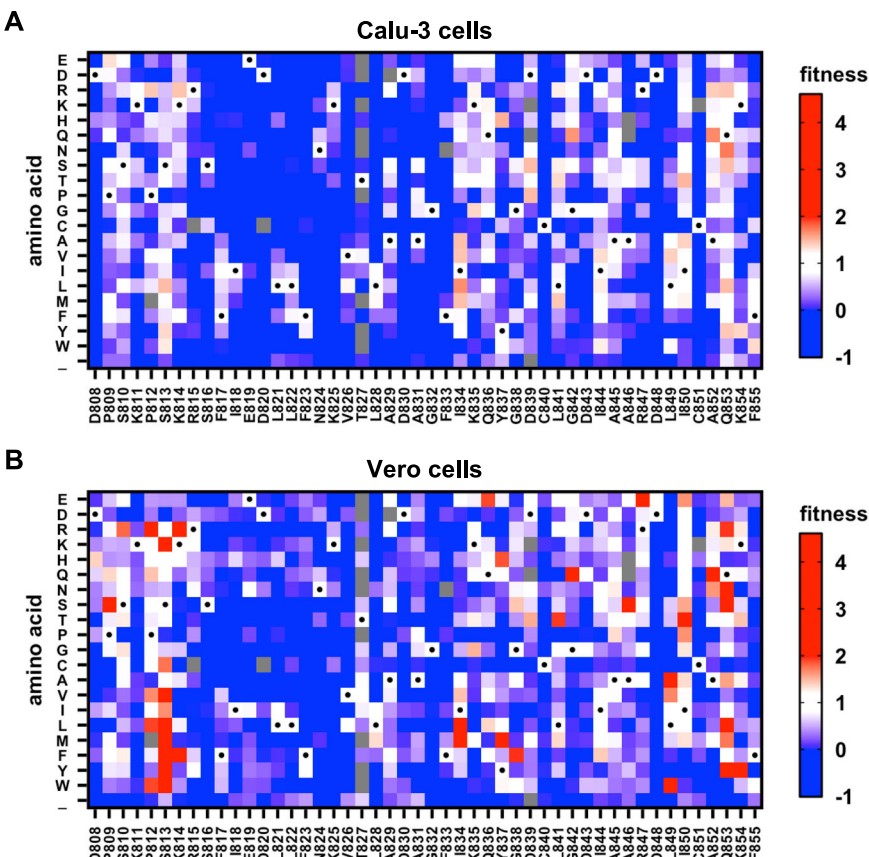

**Fig. 1 | Deep mutational scanning of SARS-CoV-2 bFP and FPPR.** The fitness values of individual mutations at residues 808–855 of SARS-CoV-2 S were measured by deep mutational scanning in **A** Calu-3 cells and **B** Vero cells and are shown as heatmaps. Wild-type (WT) amino acids are indicated by black circles. "_" indicates nonsense mutations. Mutations in gray were excluded from our data analysis due to low frequency in the plasmid mutant library. Red indicates superior fitness, white is similar to WT, and blue has reduced fitness.

S813V and S813K had reduced entry compared to WT (Fig. 2B), suggesting that mutations at residue 813 decreased sensitivity to TMPRSS2-mediated activation and hence cell surface entry. Nevertheless, camostat reduced Vero-TMPRSS2 cell entry to similar extents among WT, S813V, and S813K (Fig. S4C), indicating that they all preferred TMPRSS2-mediated entry when TMPRSS2 was overexpressed. This same experiment was also performed in the presence of fetal bovine serum (FBS), which suppresses cell surface protease-mediated (e.g., TMPRSS2-mediated) entry[45] (Fig. S4B). When FBS was added, Vero-TMPRSS2 cell entry of S813V and S813K became less sensitive to camostat, and hence exhibited less reliance on TMPRSS2, compared to WT (Fig. S4D). This observation can be explained by the efficient endosomal entry of S813V and S813K when TMPRSS2-mediated entry is highly suppressed. As a control, we also demonstrated that Calu-3 cell entry was camostat-sensitive and hence TMPRSS2-dependent (Fig. S4E), which agrees with previous studies[10,41,42]. Consistently, we observed a reduction in entry for S813V and S813K in Calu-3 cells, where TMPRSS2 is expressed on the cell surface compared to that of WT (Fig. S4F). Taken together, these results suggest that S813V and S813K have reduced TMPRSS2-mediated entry.

While S813V and S813K mutants entered Vero cells as efficiently as VSVpps with WT S proteins (Fig. 2A), they had higher fitness values than WT in the deep mutational scanning experiment (Fig. 1B). This apparent discrepancy may be explained by differences between the experimental systems. The deep mutational scanning was based on the recombinant SARS-CoV-2, whereas the VSVpp experiment only measured the efficiency of cell entry, which did not represent the entire virus life cycle. Besides, the incorporation efficiency and density of S on the virion were likely different between VSVpp and recombinant

SARS-CoV-2. Despite these differences, both the VSVpp and deep mutational scanning experiments support the hypothesis that mutations at residue 813 modulate the sensitivity to TMPRSS2-mediated activation of virus entry.

**Mutations at residue 813 decrease SARS-CoV-2 virulence in vivo**
To investigate the effects of S813V and S813K in authentic SARS-CoV-2, we introduced the two mutations individually into a mouse-adapted SARS-CoV-2 strain[38]. Vero cells, Vero-TMPRSS2 cells, and Vero cells overexpressing both TMPRSS2 and ACE2 (Vero-TMPRSS2/ACE2) were simultaneously infected with the same aliquot of the virus. The numbers of plaques obtained for WT, S813V, and S813K mutants were all enhanced in Vero-TMPRSS2 and Vero-TMPRSS2/ACE2 cells as compared to Vero cells. However, such enhancement was significantly higher for WT than S813V and S813K mutants in both Vero-TMPRSS2 and Vero-TMPRSS2/ACE2 cells (Fig. 3A, B). This observation substantiates the conclusion that S813V and S813K exhibit reduced sensitivity to TMPRSS2-mediated cleavage. Consistently, the S813V mutant also showed significantly higher titer than WT at 24 h post-infection (hpi) in Vero cells ($p = 0.01$, Fig. 3C), but not in Vero-TMPRSS2 cells (Fig. 3D).

We next aimed to understand the effects of mutations at residue 813 on virulence in mice. C57BL/6 mice were infected with 1000 or 5000 plaque-forming units (PFU) of WT, S813V, or S813K mutants. At 1000 PFU, infection with either the S813V or S813K mutant caused significantly less weight loss compared to WT (Fig. 3E). At 5000 PFU, the S813V mutant virus again resulted in less weight loss and higher survival rate than WT (Fig. 3F, G), despite having similar, if not higher virus titers in the lungs at 2 and 5 days post-infection (dpi) compared to

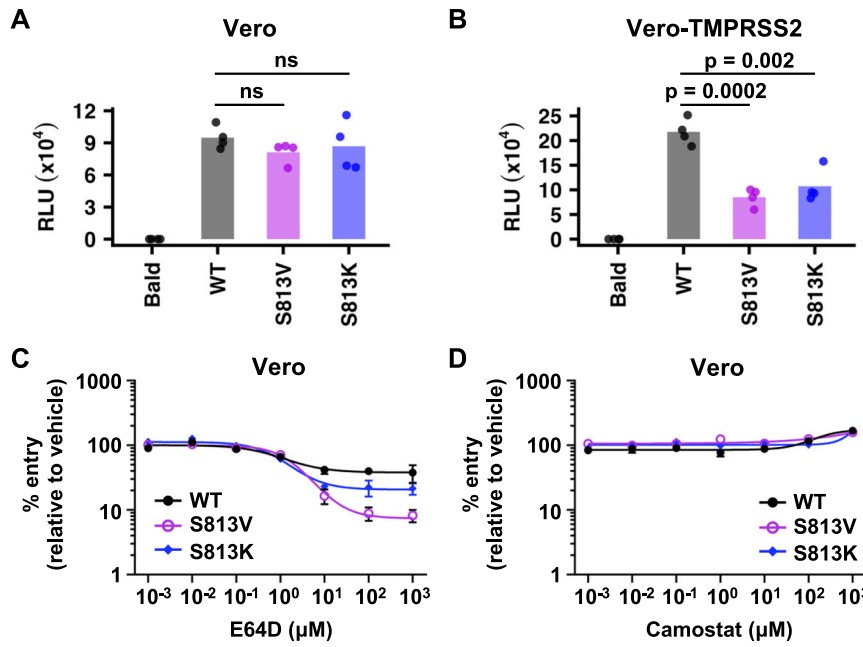

**Fig. 2 | Mutations at residue 813 influence the protease utilization during cell entry. A** Vero cell entry of VSVpps bearing various SARS-CoV-2 S constructs was measured by the relative light unit (RLU) in a luciferase assay. **B** Vero-TMPRSS2 cell entry of VSVpps bearing various SARS-CoV-2 S constructs. Each bar represents the mean of four biological replicates. Each data point represents one biological replicate. Deviations from the WT were analyzed by two-tailed *t*-tests. "ns" indicates no significance (i.e., *p*-value > 0.05). **C, D** The effects of **C** E64D (cathepsin inhibitor) or **D** camostat (TMPRSS2 inhibitor) on Vero cell entry of VSVpps bearing various SARS-CoV-2 S constructs are shown. Curves depicted in **C** are significantly different (*p* = 0.0088, two-way ANOVA). The mean and standard error of the mean (SEM) of four independent biological replicates are depicted.

WT (Fig. 3H, I). Together, these data indicate that mutations at residue 813 decreased virulence in vivo.

## Low mutational tolerance of the bFP

Although some mutations, such as those at residue 813, showed differential fitness effects between Calu-3 and Vero cells, many mutations in the deep mutational scanning experiment had consistently low fitness values between the two cell lines (Fig. 1). Subsequently, we aimed to identify regions with low mutational tolerance. Here, we defined the mutational tolerance at each residue as the average fitness value of mutations at the given residue in Calu-3 cells. Residues that interact with the host membrane should have lower mutational tolerance due to functional constraints, as demonstrated by a previous deep mutational scanning study on influenza hemagglutinin (Fig. S5)[46,47]. Notably, residues 816–833, which spanned most of the bFP, had low mutational tolerance (Fig. 4A). In contrast, the FPPR had a much higher mutational tolerance.

An NMR structure of the bFP and FPPR indicates that they form a three-helix wedge-shaped structure when interacting with the host membrane, with Leu828, which is located between helix 1 and helix 2, pointing towards the interior of the membrane[13]. Based on the mutational tolerance data, we further propose that helix 1 and the N-terminal half of helix 2, which represent the bFP, could interact with the membrane during virus-host membrane fusion. In contrast, the C-terminal half of helix 2 and all of helix 3, which represent the FPPR, would likely remain in the aqueous phase (Fig. 4B). As a result, our deep mutational scanning data substantiates that the bFP interacts with the host membrane[13,48].

We also identified three residues in the FPPR that had low mutational tolerance, namely Cys840, Asp848, and Cys851 (Fig. 4A). The low mutational tolerance of Cys840 and Cys851 could be explained by the disulfide bond between them (Fig. 4B). On the other hand, the functional importance of Asp848 was not as clear. Previous studies suggest that the bFP and FPPR each bind to a calcium ion via their negatively charged residues to promote membrane fusion[13,49]. All three negatively charged residues in the bFP, namely E819, D820, and D830, had very low mutational tolerance, consistent with these three residues representing the calcium-binding site in the bFP[49]. Our mutational tolerance data further suggested that Asp848 was the calcium-binding site in the FPPR (Fig. 4A), since it was the only negatively charged residue in the FPPR that could not tolerate any non-negatively charged mutations (Fig. 1). Consistently, Asp848, but not Asp839 and Asp843, which are the other two negatively charged residues in the FPPR, is conserved across all four genera of coronaviruses[50].

## Resistance of F823Y mutation to bFP antibodies

Previous studies have shown that antibody resistance mutations can be identified by deep mutational scanning[26,51,52]. To investigate whether SARS-CoV-2 bFP can acquire resistance mutations to bFP antibodies, deep mutational scanning was performed in the presence of bFP antibodies COV44-62 and COV44-79, both of which can neutralize SARS-CoV-2 and cross-react with coronavirus strains from different genera[17]. These two antibodies engage the bFP differently and are encoded by different germline genes[17]. COV44-62 is encoded by IGHV1-2/IGLV2-8, whereas COV44-79 is encoded by IGHV3-30/IGKV1-12[17].

Our deep mutational scanning results indicated that F823Y, which had minimal fitness cost (Fig. 1), was a resistance mutation to both COV44-62 and COV44-79 (Fig. 5A, B, Fig. S2C–F and Fig. S6A). To validate this finding, we generated VSVpp bearing SARS-CoV-2 S with the F823Y mutation. F823Y did not affect the incorporation of SARS-CoV-2 S into VSVpp, S1–S2 stability, or cleavage at the S1/S2 site (Fig. S4A). Nevertheless, F823Y S-bearing VSVpp conferred resistance to both COV44-62 and COV44-79 in a neutralization assay (Fig. 5C, D). The resistance of F823Y appeared to be stronger against COV44-79 than COV44-62 since F823Y S-bearing VSVpp was partly neutralized by COV44-62, but not COV44-79 at the highest tested concentration (500 μg/mL) (Fig. 5C, D). Consistently, the F823Y mutation weakened the binding of an epitope-containing peptide to COV44-62 and COV44-79 by 8-fold and >40-fold, respectively (Fig. S6B). These results

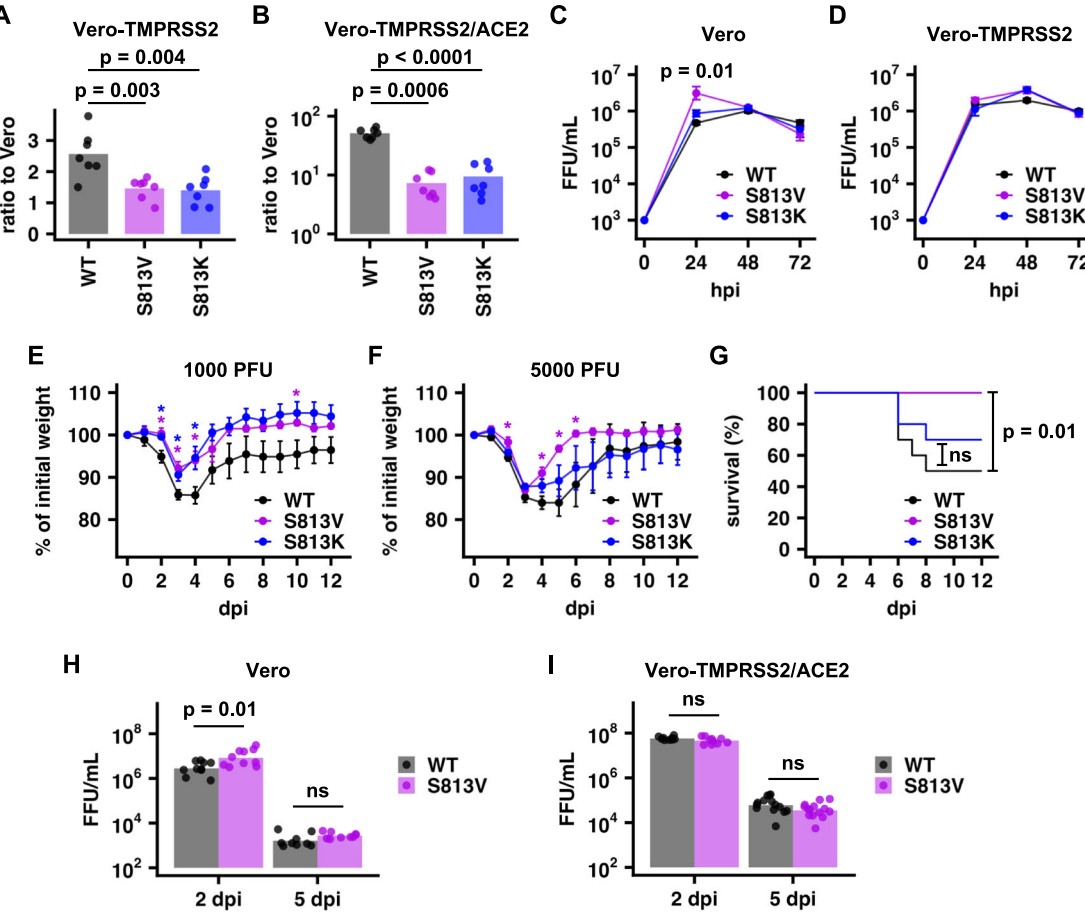

**Fig. 3 | S813V mutation reduces virulence in vivo. A, B** Vero, Vero-TMPRSS2, and Vero-TMPRSS2/ACE2 cells were separately infected with WT, S813V, or S813K viruses from the same aliquot for each virus. The numbers of plaques obtained from **A** Vero-TMPRSS2 cells or **B** Vero-TMPRSS2/ACE2 cells were normalized to those obtained from Vero cells. Bar represents the mean of seven biological replicates. Each data point represents one biological replicate. P-values were computed by two-tailed *t*-tests. **C** Vero cells or **D** Vero-TMPRSS2 cells were infected with WT, S813V, or S813K mutants at a multiplicity of infection of 0.01. Virus titers were determined for each variant at the indicated time point. Each data point represents the geometric mean of three biological replicates, and the error bar represents the geometric standard deviation (SD). Representative data from two independent experiments are shown. Deviations from the WT were analyzed by two-tailed *t*-tests. **E, F** Percentage of initial weight change of C57BL/6 mice infected with **E** 1000 PFU or **F** 5000 PFU of WT, S813V, or S813K mutants. Data points in the weight curve represent the mean, and error bars represent the SEM. Deviations from the WT were analyzed by two-tailed *t*-tests. "*" indicates *p*-value < 0.05. *p* = 0.0128, 0.00738, 0.0139, 0.0360 for WT vs. S813V at 2, 3, 4, and 10 dpi, respectively in (**E**); *p* = 0.0291, 0.0291, 0.0216 for WT vs. S813K at 2, 3, and 4 dpi respectively in (**E**). *n* = 8, 5, and 9 for WT, S813K, and S813V, respectively in (**E**); *p* = 0.0203, 0.00358, 0.00168, 0.0208 for WT vs. S813K at 2, 4, 5 and 6 dpi respectively in (**F**). *n* = 8, 10 and 9 for WT, S813K and S813V respectively in (**F**). **G** Kaplan–Meier survival curves are shown for C57BL/6 mice infected with 5000 PFU of S813V or S813K mutants. "ns" indicates not significant (i.e., *p*-value > 0.05). **H, I** Virus titers in the lungs of mice infected with 5000 PFU of WT, S813V, or S813K mutants were measured at the indicated time point on **H** Vero cells and **I** Vero-TMPRSS2/ACE2 cells. Statistical significance was determined by two-tailed t-tests. Bars represent geometric means. dpi days post-infection.

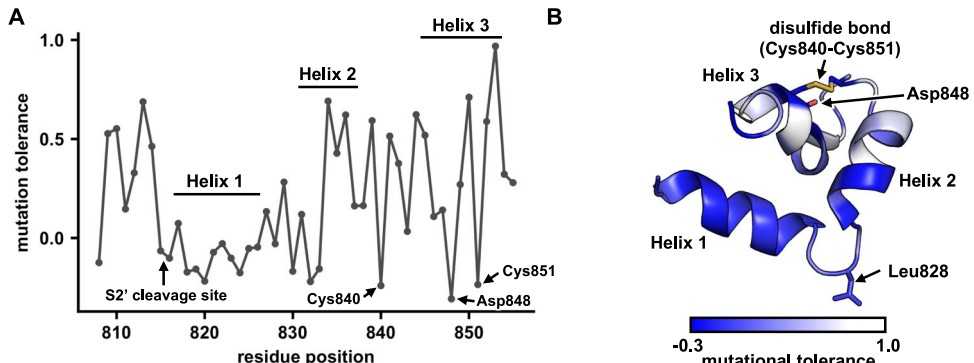

**Fig. 4 | Structural analysis of the mutational tolerance of SARS-CoV-2 bFP and FPPR. A** Mutational tolerance of each residue in Calu-3 cells is shown on the NMR structure of the bFP and FPPR (PDB 7MY8)[13]. A disulfide bond (yellow in panel **B**) is present in the FPPR between Cys840 and Cys851. **B** The mutational tolerance of each residue in Calu-3 cells is shown. The locations of helices 1–3 in the NMR structure of the bFP and FPPR (PDB 7MY8)[13] are indicated. The side chains of Leu828, Cys840, Asp848, and Cys851 are shown in stick representation.

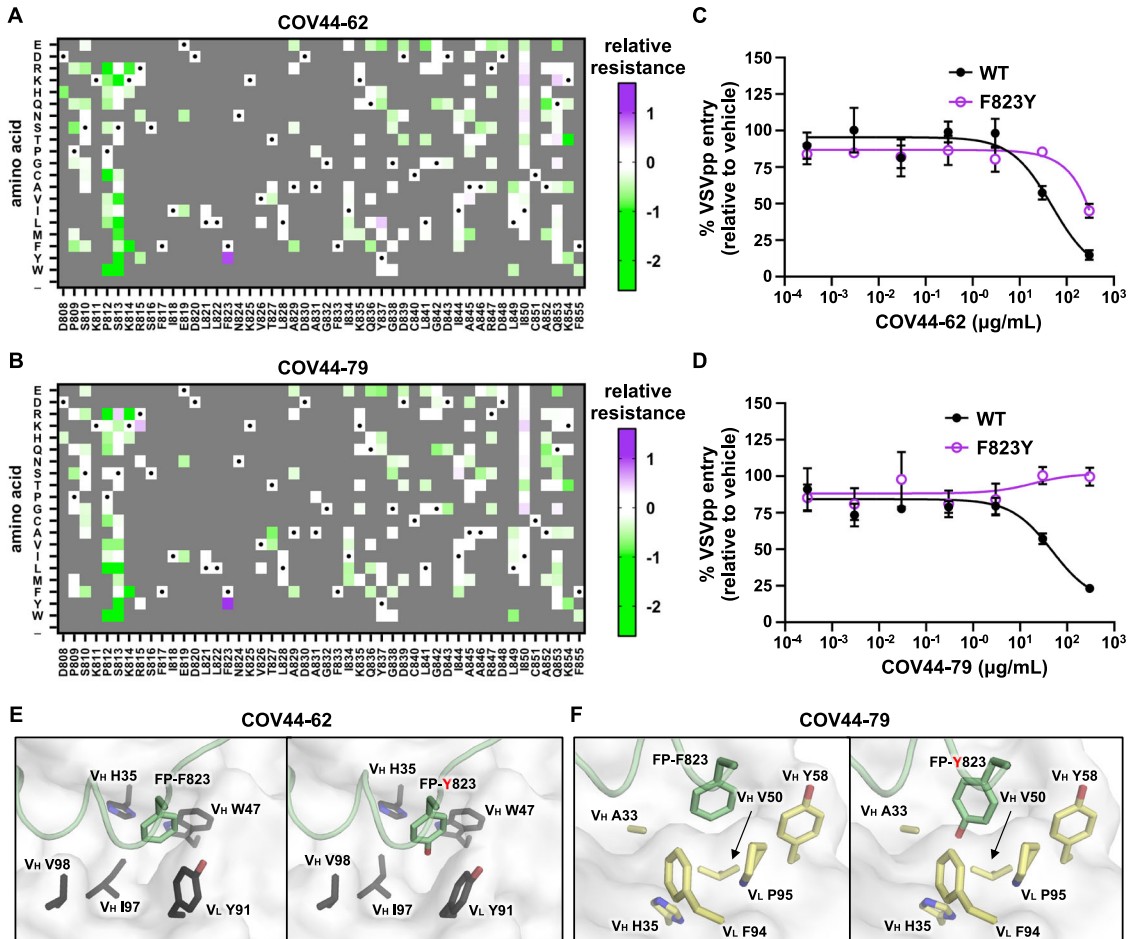

**Fig. 5 | F823Y weakens the binding of bFP antibodies. A, B** Relative resistance for each mutation against **A** 230 μg/mL COV44-62 or **B** 330 μg/mL COV44-79 in Vero cells is shown as heatmaps. Relative resistance for WT is set as 0. Mutations with a fitness value of less than 0.75 in the absence of antibodies are shown as gray. Amino acids corresponding to the WT sequence are indicated by the black dots. "_"

indicates nonsense mutations. **C, D** The neutralization activities of **C** COV44-62 and **D** COV44-79 against VSVpp bearing WT or F823Y S are shown. The mean and SEM of three biological replicates are depicted. **E, F** The structural effects of F823Y on the binding of **E** COV44-62 (PDB 8D36)[17] and **F** COV44-79 (PDB 8DAO)[17] were modeled using FoldX[53].

demonstrate that resistance to bFP antibodies can be conferred by a single mutation.

To understand the structural mechanism of antibody resistance, we further analyzed the previously determined x-ray crystal structures of COV44-62 and COV44-79 in complex with SARS-CoV-2 bFP[17]. FoldX was used to model the structural effect of F823Y mutation[53]. The major difference between Phe and Tyr is an extra side-chain hydroxyl group on Tyr. Our models showed that the hydroxyl group of Tyr823 pointed towards the bottom of hydrophobic pockets formed in the COV44-62 and COV44-79 binding sites (Fig. 5E, F). Burying a polar hydroxyl group on the Tyr side chain without forming any H-bond would impose an appreciable desolvation energy cost[54]. Consistently, FoldX indicated that the F823Y mutation weakened the binding energy of COV44-62 and COV44-79 by 1.0 and 1.2 kcal/mol, respectively. These observations provide a mechanistic basis for the resistance to bFP antibodies conferred by F823Y.

## Discussion

Most studies of the SARS-CoV-2 S protein focus on the RBD since it is immunodominant and engages the host receptor ACE2 for cell entry[1,22,23]. In contrast, the S2 domain is less well characterized. The bFP and its flanking regions were chosen for our study here as their role in mediating the fusion process has not been thoroughly characterized. Since no antivirals are currently approved by the Food and Drug Administration to target this region, mutations within this region

should not inhibit the efficacy of any clinical regimen available for treating SARS-CoV-2 infection. Furthermore, mutagenesis described in this study was performed in the S2 region instead of the immunodominant RBD in the S1 subunit. As a result, most people should have robust immunity against all the SARS-CoV-2 mutants in our library due to vaccinations or prior infections. Our study here provides important insights into how mutations in the regions adjacent to the S2' cleavage site can modulate the preference of cell entry pathway as well as promote resistance to broadly neutralizing antibodies, which may inform vaccine design to prevent future coronavirus pandemics. Our results also advance the knowledge of the evolutionary potential of the SARS-CoV-2 S2 domain and demonstrate the feasibility of applying deep mutational scanning to authentic SARS-CoV-2.

A key result in this study is the low mutational tolerance of the bFP, which substantiates its functional importance during membrane fusion[13–15]. However, the bFP was located outside the membrane as a disordered region in a recent cryo-EM structure of postfusion SARS-CoV-2 S in a lipid bilayer membrane[16]. Instead, the iFP inserts into the membrane in this cryo-EM structure. However, the postfusion SARS-CoV-2 S in this cryo-EM structure does not have a cleaved S2' site that is essential for membrane fusion during virus entry[16,55]. Besides, this cryo-EM structure was determined at pH 7.5 without any calcium ions, while SARS-CoV-2 S-mediated membrane fusion requires an acidic pH[56] and the presence of calcium ions[49,57]. While it is always challenging for structural and biophysical studies of viral fusion proteins to emulate

the physiological states as would occur in vivo, it is possible that both bFP and iFP of SARS-CoV-2 S interact with the host membrane[13,16], but at different stages of the membrane fusion process. Future studies are therefore needed to better characterize the molecular mechanisms of the highly dynamic S-mediated membrane fusion process.

Another major observation in our study is that mutations at SARS-CoV-2 S residue 813 influenced host cell entry and sensitivity to TMPRSS2-mediated S2′ cleavage. Consistently, similar findings on residue 813 have recently been described for different SARS-CoV-2 variants as well as SARS-CoV[58]. Previous studies showed that H655Y and N969K mutations in Omicron can shift the preference from TMPRSS2-mediated cell surface entry to cathepsins-mediated endosomal entry, resulting in reduced virulence[8-10]. The proposed underlying mechanism is that they stabilize the S trimer and hence decrease the fusogenicity and cell surface entry efficiency[9-11]. We also observed this relationship between the cell entry pathway and virulence in mutations at residue 813. However, unlike H655Y and N969K, residue 813 is near the S2′ cleavage site (Fig. S7). Therefore, while S813V and S813K have similar phenotypes as H655Y and N969K, their molecular mechanisms are unlikely to be the same. Given that other residues flanking the S2′ cleavage site can also modulate protease preference for S2′ cleavage[59,60], mutations in this region may provide valuable information on the preference of cell entry pathway and pathogenicity as SARS-CoV-2 continues to evolve. At the same time, mutations with differential fitness effects between Calu-3 and Vero cells that are distal from the S2′ cleavage site may represent mechanisms unrelated to S2′ cleavage. We believe those mutations, albeit outside of the scope of this study, warrant future investigations.

There are currently five coronavirus strains circulating in the human population (229E, HKU1, NL63, OC43, and SARS-CoV-2). In addition, other zoonotic coronaviruses continue to pose a pandemic threat[61]. As a result, developing a pan-coronavirus vaccine has become an attractive idea, especially after the discovery of broadly neutralizing antibodies to the bFP[17,20,21,24]. However, despite the high sequence conservation of the bFP, our study here found that the F823Y mutation can confer strong resistance against bFP antibodies. F823Y is a natural variant in bat betacoronavirus HKU9 and is also observed in circulating SARS-CoV-2 at a very low frequency (Fig. S8). Although these observations represent a potential obstacle to the development of a pan-coronavirus vaccine, resistance mutations against bFP antibodies are rare in our deep mutational scanning results, partly due to the high fitness cost of most mutations in the bFP. Therefore, we concur that the bFP is a promising target for the development of a pan-coronavirus vaccine[17,20,21,24].

## Methods

### Cell lines and antibodies
HEK293T (ATCC, CRL-3216), Calu-3 (BEI resources, NR-55340), Vero (ATCC, CRL-1586), Vero-TMPRSS2 (generated by Dr. Michael Diamond, Washington University in St. Louis) and Vero-TMPRSS2/ACE2 (BEI resources, NR-54970) cells were maintained in Dulbecco's Modified Eagle Media (DMEM) containing 10 mM HEPES, 100 nM sodium pyruvate, 0.1 mM non-essential amino acids, 100 U/ml penicillin G, and 100 μg/ml streptomycin, and supplemented with 10% fetal bovine serum (FBS, Atlanta Biologicals). Calu-3 cells were maintained in Minimum Essential Media (MEM) supplemented with 20% FBS, 100 U/ml penicillin G, and 100 μg/ml streptomycin. All cell lines were cultured in a 5% $CO_2$ incubator at 37 °C. Antibodies used in this study were purchased from commercial vendors as listed: SARS-CoV-2 anti-N antibody (SinoBiological, Cat. #: Cat: 40143-R001); CoV 44-79 and CoV 44-62 were expressed and purified in-house as described below. Rabbit polyclonal anti-SARS-CoV-2-S1 (SinoBiological, catalog #: 40591-T62); Mouse anti-C9 (EMD Millipore, catalog #: MAB5356); Mouse monoclonal anti-VSV-M (KeraFast, catalog #: EB0011); HRP anti-mouse antibody (Thermo Fisher, catalog #: 31430); HRP anti-rabbit (Thermo

Fisher, catalog #: 31460). Antibodies purchased from commercial vendors have been validated by the manufacturer for the specific application in this study. Experiments were performed under the conditions specified by the manufacturer.

### SARS-CoV-2 infection of mice
All experiments with SARS-CoV-2 were performed in a biosafety level 3 (BSL3) laboratory at the University of Iowa. All animal studies were approved by the University of Iowa Animal Care and Use Committee and meet stipulations of the Guide for the Care and Use of Laboratory Animals (protocol # 2071795-013). C57BL/6 mice of both sexes at 4–6 months old were purchased from NCI, Charles Rivers. Mice were housed under standard conditions of dark/light cycle, ambient temperature, and humidity. Mice were anesthetized with ketamine-xylazine and infected intranasally with the indicated amount of virus in a total volume of 50 μL DMEM. Animal weight and health were monitored daily.

### Virus titer by plaque assay
At the indicated times, mice were euthanized and transcardially perfused with PBS. Lungs were collected and homogenized before clarification by centrifugation and tittering. Virus or tissue homogenate supernatants were serially diluted in DMEM. Vero, Vero-TMPRSS2, or Vero-TMPRSS2/ACE2 cells in 12-well plates were inoculated at 37 °C in 5% $CO_2$ for 1 h and gently rocked every 15 min. After removing the inocula, plates were overlaid with 0.6% agarose containing 2% FBS. After 3 days, overlays were removed, and plaques were visualized by staining with 0.1% crystal violet. Viral titers were quantified as PFU per mL tissue.

### Virus titer by focus forming assay
Virus or tissue homogenate supernatants were serially diluted in DMEM. Vero, Vero-TMPRSS2, or Vero-TMPRSS2/ACE2 cells in 96-well plates were inoculated at 37 °C in 5% $CO_2$ for 1 h and gently rocked every 15 min. After removing the inocula, plates were overlaid with 1.2% methylcellulose containing 2% FBS. The next day, overlays were removed, and cells were stained with anti-nucleocapsid antibody for SARS-CoV-2 (1:1000) for 1 h at 37 °C and then with HPR-conjugated secondary antibody for 1 h at 37 °C. Foci were visualized by peroxidase substrate. Viral titers were quantified as fluorescent focus unit (FFU) per mL tissue.

### Virus growth assay
Vero or Vero-TMPRSS2 cells in 12-well plates were infected with 0.01 MOI of the indicated virus diluted in DMEM. Cells were frozen at the indicated time points. Virus titers were determined by either plaque assay or focus forming assay. Three biological replicates were included for each time point.

### Mutant library construction
A mutant library of residues 808-855 of SARS-CoV-2 S was constructed based on a BAC-based reverse genetic system of SARS-CoV-2 Wuhan-Hu-1 (p-BAC SARS-CoV-2)[37,38]. Saturation mutagenesis was performed using an overlapping PCR strategy as described previously[32]. Briefly, a library of mutant inserts was generated by two separate batches of PCRs to cover the entire region of interest (residues 808–855). The first batch of PCRs consisted of 6 reactions, each containing one cassette of forward primers and the universal reverse primer 5′- GCC AAT AGC ACT ATT AAA TTG GTT-3′. Each cassette contained an equal molar ratio of eight forward primers that had the same 21 nucleotides (nt) at the 5′ end and 15 nt at the 3′ end. Each primer within a cassette was also encoded with an NNK (N: A, C, G, T; K: G, T) sequence at a specified codon position for saturation mutagenesis. In addition, each primer also carried unique silent mutations (also known as synonymous mutations) to help distinguish between sequencing errors and true

mutations in downstream sequencing data analysis as described previously[62]. The forward primers, named CassetteX_N (X: cassette number, N: primer number), are listed in Table S1. The second batch of PCR consisted of another 6 PCRs, each with a universal forward primer 5'-ATG TAC ATT TGT GGT GAT TCA ACT-3' and a unique reverse primer as listed in Table S1. Subsequently, 6 overlapping PCRs were performed using the universal forward and reverse primers, as well as a mixture of 10 ng each of the corresponding products from the first and second batches of PCR. The 6 overlap PCR products were then mixed at equal molar ratios to generate the final insert of the mutant library. All PCRs were performed using PrimeSTAR Max polymerase (Takara Bio, catalog no. R045B) per the manufacturer's instruction, followed by purification using the Monarch Gel Extraction Kit (New England Biolabs, catalog no. T1020L).

The mutant library PCR product (residues 808–855) was introduced into the SARS-CoV-2 BAC encoding Wuhan-Hu-1 sequence by a two-step linear lambda red recombination process[63,64]. The first step removed and replaced the region of interest with the GalK-Kan selection marker, while the second step removed and replaced the GalK-Kan selection marker with the mutant library PCR product. In brief, the GalK-Kan selection marker flanked by the SARS-CoV-2 sequence was PCR-amplified from pYD-C225[63] and gel-purified. Gel-purified GalK-Kan fragments were transformed into SW102 cells carrying the SARS-CoV-2 BAC by electroporation for linear lambda red recombination. Recombinants were selected by Kanamycin resistance culture plates. The presence of GalK-Kan cassette in selected recombinants was verified by PCR with primers flanking the area of recombination: 5'-CCA TAC CCA CAA ATT TTA CTA TTA GTG TTA CCA CA-3' and 5'-TTG ACC ACA TCT TGA AGT TTT CCA AGT G-3'). Verified recombinants were further introduced with the mutant library PCR product (residues 808–855) by electroporation for a second round of linear lambda red recombination. Two electroporation was performed separately to obtain two independent BAC mutant libraries as replicates. Successful recombinants were selected using 2-deoxy-galactose-based culture plates. All viable clones were collected and pooled to generate the BAC mutant library. The loss of the GalK-Kan cassette (and hence the SARS-CoV-2 sequence) in the BAC mutant library was confirmed by PCR with primers flanking the area of recombination: 5'-CCA TAC CCA CAA ATT TTA CTA TTA GTG TTA CCA CA-3' and 5'-TTG ACC ACA TCT TGA AGT TTT CCA AGT G-3'. GalK-Kan selection markers were amplified with primers: 5'-ATG TAC ATT TGT GGT GAT TCA ACT GAA TGC AGC AAT CTT TTG TTG CAA TAC CTG TTG ACA ATT AAT CAT CG-3' and 5'-GCC AAT AGC ACT ATT AAA TTG GTT GGC AAT CAA TTT TTG TTC TCA TAG ACT CAG CAA AGT TCG ATT TA-3'. Sequences complementary to pYD-C225 are underlined.

S813V and S813K were first individually introduced to an expression construct encoding SARS-CoV-2 S with an NEB Q5 site-directed mutagenesis kit. S813K was introduced with primers: 5'-ATC AAA ACC AAA GAA GAG GTC ATT TAT TG-3' and 5'- GGA TCT GGT AAT ATT TGT G-3'; S813V was introduced with primers: 5'-ATC AAA ACC AGT GAA GAG GTC ATT TAT TGA AG-3' and 5'- GGA TCT GGT AAT ATT TGT G-3'. The mutated codons for S813K and S813V are underlined. The part of the S protein encoding S813K or S813V were separately amplified with primers: 5'-CCA TAC CCA CAA ATT TTA CTA TTA GTG TTA CCA CA-3' and 5'-TTG ACC ACA TCT TGA AGT TTT CCA AGT G-3' from the expression construct of the SARS-CoV-2 S encoding S813K or S813V generated from the site-directed mutagenesis process. The PCR products were introduced into SARS-CoV-2 BAC as described above.

### Rescue and passage of the viral mutant library

2 μg of BAC mutant library were transfected into Vero cells with Lipofectamine 3000 (Thermo Fisher Scientific, catalog #: L3000008) into each well of a 6-well plate according to the manufacturer's protocol (12 μg in total for each replicate). Cells were monitored daily for cytopathic effects (CPE). Cultures were harvested when CPE was >50%

by freezing at −80 °C. Viruses rescued from each well of the transfected 6-well plate were pooled independently for each replicate to generate the P0 virus. The titers for the P0 virus were determined by plaque assay and further passaged in Calu-3 or Vero cells at an MOI of 0.01 in DMEM supplemented with 10% FBS. P1 viruses were harvested at 48 h post-infection by freezing at −80 °C. SARS-CoV-2 BAC with S813K or S813V mutations were recovered as described above.

For the antibody resistance selection, bFP antibodies were incubated with the P0 viruses at a concentration that corresponds to PRNT$_{90}$ at 37 °C for 1 h. The amount of P0 viruses used corresponds to the amount needed for infection at an MOI of 0.01 in a T75 flask. Calu-3 or Vero cells were then infected with the virus inoculum for 1 h in the presence of 230 μg/mL COV44-62 antibody or 330 μg/mL COV44-79 antibody. The virus inoculum was removed after virus adsorption, and cells were washed with PBS before supplementing the culture medium with 230 μg/mL COV44-62 antibody or 330 μg/mL COV44-79 antibody. Supernatant and cells were harvested at 48 h post-infection by freezing at −80 °C.

### Sequencing library preparation

Viruses from different passages were inactivated in TRIzol (Thermo Fisher Scientific, catalog no. 15596026) for RNA isolation as specified by the manufacturer's protocol. Isolated RNA was subject to DNase I treatment (Thermo Fisher Scientific, catalog no. 18068015) and reverse-transcribed using the SuperScript IV First-Strand Synthesis System with random hexamers (Thermo Fisher Scientific, catalog no. 18091050). The region corresponding to residues 805-864 was amplified from the cDNA (post-selection) or the BAC mutant library (input) using KOD Hot Start DNA polymerase (MilliporeSigma, catalog no. 710863) per the manufacturer's instruction with the following two primers: 5'-CAC TCT TTC CCT ACA CGA CGC TCT TCC GAT CTT TTG GTG GTT TTA ATT TTT CAC AA-3' and 5'-GAC TGG AGT TCA GAC GTG TGC TCT TCC GAT CTT TGA GCA ATC ATT TCA TCT GTG AG-3'. Sequences complementary to the cDNA are underlined, whereas the rest of the sequences correspond to the Illumina adapter sequence. An additional PCR was performed to add the rest of the Illumina adapter sequence and index to the amplicon using primers: 5'-AAT GAT ACG GCG ACC ACC GAG ATC TAC ACX XXX XXX XAC ACT CTT TCC CTA CAC GAC GCT-3' and 5'-CAA GCA GAA GAC GGC ATA CGA GAT XXX XXX XXG TGA CTG GAG TTC AGA CGT GTG CT-3'. Positions annotated by an X represent the nucleotides for the index sequence. The final PCR products were purified by PureLink PCR purification kit (Thermo Fisher Scientific, catalog no. K310002) and submitted for next-generation sequencing using Illumina MiSeq PE250.

### Sequencing data analysis

Next-generation sequencing data were obtained in FASTQ format. Forward and reverse reads of each paired-end read were merged by PEAR[65]. The merged reads were parsed by SeqIO module in BioPython[66]. Primer sequences were trimmed from the merged reads. Trimmed reads with lengths inconsistent with the expected length were discarded. The trimmed reads were then translated to amino acid sequences, with sequencing error correction performed at the same time as previously described[62]. Amino acid mutations were called by comparing the translated reads to the WT amino acid sequence. The frequency (F) of a mutant i within sample s of replicate k was computed for each replicate as follows:

$$F_{i,s,k} = \frac{\text{readcount}_{i,s,k} + 1}{\sum_i (\text{readcount}_{i,s,n,k} + 1)} \tag{1}$$

Mutants with a frequency of <0.01% in the BAC mutant library were discarded.

Enrichment score (ES) of mutant $i$ in replicate $k$ was calculated as follows:

$$ES_{i,k} = \log_{10} \frac{F_{i,k,\text{post-selection}}}{F_{i,k,\text{input}}} \quad (2)$$

Fitness value ($W$) of a mutant $i$ in replicate $k$ was calculated as follows:

$$W_{i,k} = \frac{ES_{i,k} - \overline{ES_{\text{nonsense},k}}}{\overline{ES_{\text{silent},k}} - \overline{ES_{\text{nonsense},k}}} \quad (3)$$

where $\overline{ES_{\text{silent},k}}$ and $\overline{ES_{\text{nonsense},k}}$ represent the average ES for silent and nonsense mutations, respectively, in replicate $k$.

The final fitness value for each mutant was the average $W$ of the two replicates. The mutational tolerance for each residue was computed as the average fitness value of mutations at the given residue.

Relative resistance ($R$) for a given mutant i against antibody $a$ in cell line $c$ was computed as follows:

$$R_{i,c,a} = W_{i,c,a} - W_{i,c,\text{no antibody}} \quad (4)$$

## Sanger sequencing of individual clones
The two BAC mutant libraries were used as templates for amplifying spike residues 808-855 with the following forward primer encoding EcoRI site: 5′-GGT ACC GAA TTC CCA TAC CCA CAA ATT TTA CTA TTA GTG TTA CCA CA-3′ and reverse primer encoding NotI site:5′- GGT ACC GCG GCC GCT TGA CCA CAT CTT GAA GTT TTC CAA GTG-3′. EcoRI and NotI sequences are underlined in the forward and reverse primer, respectively. The PCR products were purified with PureLink™ Quick Gel Extraction Kit (Thermo Scientific) and digested with EcoRI and NotI (New England Biolabs). The digested DNA products were purified and ligated with EcoRI/NotI digested pcDNA3.1 vector with T4 DNA Ligase (New England Biolabs). Individual clones were selected and DNA was isolated with QIAprep Spin Miniprep Kit (Qiagen). Purified DNA was sequenced with the following primer: 5′-CCA TAC CCA CAA ATT TTA CTA TTA GTG TTA CCA CA-3′.

## Antibody expression and purification
The heavy chain and light chain of the indicated antibodies were cloned into phCMV3 plasmids in an IgG1 or Fab format with a mouse immunoglobulin kappa signal peptide. Plasmids encoding the heavy chain and light chain of antibodies were transfected into Expi293F cells using an ExpiFectamine 293 transfection kit (Gibco) in a 2:1 mass ratio following the manufacturer's protocol. The supernatant was harvested 6 days post-transfection and centrifuged at $4000 \times g$ for 30 min at 4 °C to remove cells and debris. The supernatant was subsequently clarified using a polyethersulfone membrane filter with a 0.22 μm pore size (Millipore).

CaptureSelect CH1-XL beads (Thermo Scientific) were washed with MilliQ $H_2O$ thrice and resuspended in 1× PBS. The clarified supernatant was incubated with washed beads overnight at 4°C with gentle rocking. Then, flowthrough was collected, and beads were washed once with 1× PBS. Beads were incubated in 60 mM sodium acetate, pH 3.7, for 10 min at 4 °C. The eluate-containing antibody was buffer-exchanged into 1× PBS and further purified by size-exclusion chromatography using Superdex 200 XK 16/100 column in 1× PBS. Antibodies were stored at 4 °C.

## Biolayer interferometry binding assay
Binding assays were performed by biolayer interferometry (BLI) using an Octet Red instrument (FortéBio). Briefly, an N-terminally biotinylated peptide of SARS-CoV-2 S (808-DPSKPSKRSFIEDLLFNKVT-827) as well as a version with F823Y mutation at 50 μg/ml in 1× kinetics buffer (1× PBS, pH 7.4, 0.01% BSA and 0.002% Tween 20) were loaded onto SA biosensors and incubated with the COV44-62 and COV44-79 Fabs at 33.3 nM, 100 nM, and 300 nM. The assay consisted of five steps: (1) baseline: 60 s with 1× kinetics buffer; (2) loading: 180 s with biotinylated peptides, (3) baseline: 60 s with 1× kinetics buffer; (4) association: 180 s with Fabs; and (5) dissociation: 180 s with 1× kinetics buffer. For estimating the exact KD, a 1:1 binding model was used.

## Pseudovirus virus entry assay
Full-length SARS-CoV-2 S gene (GenBank: NC_045512.2) was synthesized by GenScript. as human codon-optimized cDNAs, and inserted into pcDNA3.1 expression vector[67]. C9-tagged versions of the S genes were generated by replacing the 3′-terminal 19 codons with linker and C9 codons (GSSGGSSG-GGTETSQVAPA)[68]. All S recombinants were constructed via gene fragment Assembly (New England Biolabs, catalog #: E2621S).

pHEF-VSVG-Indiana was constructed previously[69]. VSVGΔG-fluc-G pseudoviral particles (VSVpps[70]) stock was made as previously described[71]. Briefly, HEK293T cells were transfected with VSV-G. The next day, seed VSVΔG-G particles were inoculated onto the transfected cells for 2 h. The cells were rinsed three times with an FBS-free DMEM medium and replenished with fresh media. After a 48-h incubation period, media were collected and clarified ($300 \times g$, 4 °C, 10 min then $3000 \times g$, 4 °C, 10 min). To obtain purified viral particles, clarified VLP-containing media were laid on top of 20% w/w sucrose cushions, and viral particles were purified via slow-speed pelleting (SW28, 6500 rpm, 4 °C, 24 h). The resulting pellet was resuspended in FBS-free DMEM to 1/100 of the original volumes. Concentrated particle stocks were stored at −80 °C until used.

VSVpps bearing various recombinant SARS-CoV-2 S proteins were used to infect different cell types. VSVpps were quantified based on VSV-M expression by Western Blot analysis. For protease/antibody inhibition experiments, cells were pre-incubated with serial dilutions of camostat, E64D, or antibodies for 1 h at 37 °C before VSVpp inoculation. Inoculation was allowed to infect cells for 2 h, then cells were rinsed 3 times and replenished with cell culture media (with 10% FBS). Following overnight incubation, cells were lysed by lysis buffer (25 mM Tris-phosphate pH 7.8, 2 mM dithiothreitol, 2 mM 1,2-diaminocyclo-hexane-$N,N,N'$-tetraacetic acid, 10% glycerol, 1% Triton X-100). Firefly luciferase (VSVpp) activity was recorded by a Veritas microplate luminometer by the addition of substrate (1 mM d-luciferin, 3 mM ATP, 15 mM $MgSO_4 \cdot H_2O$, 30 mM HEPES pH 7.8).

## Western blot analysis
Samples in SDS solubilizer (0.0625 M Tris·HCl pH 6.8, 10% glycerol, 0.01% bromophenol blue, 2% SDS, and 2% 2-mercaptoethanol) were heated at 95 °C for 5 min, electrophoresed through 8% polyacrylamide-SDS gels, transferred to nitrocellulose membranes (Bio-Rad), and incubated with rabbit polyclonal anti-SARS-CoV-2-S1 (SinoBiological, catalog #: 40591-T62; 1:1000), mouse anti-C9 (EMD Millipore, catalog #: MAB5356; 1:1000), mouse monoclonal anti-VSV-M (KeraFast, catalog #: EB0011; 1:1000). After incubation with anti-mouse (Thermo Fisher, catalog #: 31430, 1:5000) or anti-rabbit (Thermo Fisher, catalog #: 31460, 1:5000) HRP-tagged secondary antibodies and chemiluminescent substrate, or purified LgBiT-substrate cocktail (Promega), the blots were imaged and processed with a FluorChem E (Protein Simple).

## Structural modeling
FoldX[53] was used to model the structural and protein stability effects of mutation F823Y. The published structures of SARS-CoV-2 bFP in complex COV44-62 (PDB 8D36)[17] and COV44-79 (PDB 8DAO)[17] were used as input.

## Sequence alignment

Sequence alignment was performed using (http://www.bioinformatics.org/sms/multi_align.html)[72]. Sequences were downloaded from NCBI GenBank database (www.ncbi.nlm.nih.gov/genbank)[73]. Genbank IDs for the S sequences used are as follows:

ABB90529.1: Human coronavirus 229E (HCoV-229E)

YP_003767.1: Human coronavirus NL63 (HCoV-NL63)

ADN03339.1: Human coronavirus HKU1 (HCoV-HKU1)

AIX10756.1: Human coronavirus OC43 (HCoV-OC43)

YP_001039971.1: Rousettus bat coronavirus HKU9 (Bat-CoV-HKU9)

ABF65836.1: Severe acute respiratory syndrome-related coronavirus (SARS-CoV)

QHD43416.1: Severe acute respiratory syndrome coronavirus 2 (SARS-CoV-2)

AHX00731.1: Middle East respiratory syndrome-related coronavirus (MERS-CoV)

YP_001876437.1: Beluga whale coronavirus SW1 (BWCoV-SW1)

AHB63508.1: Bottlenose dolphin coronavirus HKU22 (BDCoV-HKU22)

AFD29226.1: Night heron coronavirus HKU19 (NHCoV-HKU19)

AFD29187.1: Porcine coronavirus HKU15 (PDCoV-HKU15)

## Reporting summary

Further information on research design is available in the Nature Portfolio Reporting Summary linked to this article.

## Data availability

Raw sequencing data have been submitted to the NIH Short Read Archive under accession number: BioProject PRJNA910585. NMR structure of the bFP and FPPR were retrieved from PDB (7MY8). Raw data are provided in the "Source Data" file with this paper. Source data are provided in this paper.

## Code availability

Custom Python scripts for all analyses have been deposited to: https://zenodo.org/records/10841475[74].

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

## Acknowledgements

This work was supported by National Institutes of Health (NIH) R21 AI178391 (T.G.), R00 AI170996 (L.-Y.R.W.), P01 AI060699 (S.P.), R01 AI129269 (S.P.), the Searle Scholars Program (N.C.W.) and the Bill and Melinda Gates Foundation grant INV-004923 (I.A.W.).

## Author contributions

R.L., E.Q., T.G., S.P., N.C.W., and L.-Y.R.W. conceived and designed the study. L.-Y.R.W., R.L., A.O., and C.D.G. performed the deep mutational scanning experiments. N.C.W. and N.T.Y.S analyzed the deep mutational scanning data. R.L., T.J.C.T., and W.O.O. expressed and purified the antibodies. E.Q. performed the functional characterization experiments. M.Y. and I.A.W. performed the biolayer interferometry experiment. L.-Y.R.W. and N.C.W. wrote the paper and all authors reviewed and/or edited the paper.

## Competing interests

N.C.W. consults for HeliXon. The authors declare no competing interests.
