## [Peer Review File · Nature Communications]

Functional and antigenic characterization of SARS-CoV-2 spike fusion peptide by deep mutational scanningREVIEWER COMMENTS

Reviewer #1 (Remarks to the Author):

Lei, Qing et al. report deep mutational scanning studies on the SARS-CoV-2 spike protein focused on the fusion peptide and adjacent region. In replicating Wuhan-Hu-1 virus generated from reverse genetics, they constructed a mutant library encompassing residues 808-855 which includes the bona fide fusion peptide (816-834), the S2' cleavage site (815/816), and fusion peptide proximal region (but not a putative internal fusion peptide suggested from a recent structure, 867-909). Mutational constraint confirms the importance of the bona fide fusion peptide. They identify mutations with protease-specific effects on viral replication fitness and mutations escaping broadly cross-reactive FP mAbs. This study is to my knowledge the first deep mutational scanning experiment in a fully replicating SARS-CoV-2 virion as opposed to pseudovirus or cell-display platforms, which is a big innovation. The conclusions are interesting and validated nicely. I find this to be a nice and well-performed study.

Minor comments, in no particular order:

1. Have mutations at position 813 occurred at all during the pandemic, either in named variants or in other sequences seen during viral surveillance (e.g. chronic infection lineages). What about other mutations/sites focused on during this scan?
2. Figure 1: “*” could be used instead of “-” to denote nonsense mutations (“-” would typically represent a gap character i.e. deletion)
3. Description of biosafety considerations and cost/benefit analysis of creating replicating mutant SARS-CoV-2 should be considered.

Reviewer #1 (Remarks on code availability):

link does provide a README with all scripts and appears to be complete, though I did not attempt to install or run the code myself

Reviewer #2 (Remarks to the Author):

Lei and colleagues significantly advance our understanding of the effects of mutations in the SARS-CoV-2 spike protein in this paper. They employ deep mutational scanning methods to characterize and pinpoint crucial residues within the SARS-CoV-2 spike protein, utilizing in vivo assays that can be categorized into two distinct sections. One noteworthy finding in the manuscript highlights the relationship between cell tropism and TMPRSS2 cleavage events, emphasizing the role of the S813 residue. Conversely, the F823 mutation is identified as impeding the binding of antibodies. While the manuscript is well-written and represents substantial work, some pertinent issues need to be

addressed at this stage. This mainly clarifies and elaborates on novel findings about S813 and the future direction of the discussion.

1. Importance of S813 residue and entry in different cell types

Based on the hypothesis, the Vero cell prefers entering through the endosome, while Calu-3 prefers entering through the plasma membrane because the Vero cell lacks TMPRSS2. Testing the hypothesis of TMPRSS2 overexpression and membrane fusion pathway is very confusing since there is so much data to digest (paragraph lines 134-150). Consider polishing the section to help readers to understand better.

- The data show many Vero data assays but only one with Calu-3 in the supplement. It would be beneficial to show Calu-3 data in detail to understand better. I wonder if we should see a reduction in mutants compared to WT since S813 mutants have reduced entry in the presence of TMPRSS2. Figure S2E only shows a relative % entry. Also, what happens when you add FBS to it? Will it reduce entry, as shown in the Vero cell line?

- In the entry assay, Were VSVpps added based on the protein concentration measured in the western blot? How do you quantify the amount of VSVpps? It should be mentioned since the expression levels of the WT and mutants are different.

- In Figure 2, the entry assay in Vero-TMPRSS2 cells(B) shows that S813V and S813K both reduce but still have similar readings to Vero cells(A). Does this suggest that VSVpps will take both pathways when TMPRSS2 is expressed? It is unclear whether the plasma membrane fusion pathway is preferred or if they still have both pathways take one or another. Please clarify this.

- Please explain and discuss all of the data presented such as the discussion about Figure S2D will be helpful. I wonder if it is due to less sensitivity or more endosomal fusion.

- If it is possible, it might be interesting to see if you can correlate Figure 3 data with Figure 2.

2. Low mutational tolerance of the bFP

- Line 185, showed differential fitness effects between Calu-3 and Vero cells. I am not clear other than the heat map. Is there any other effect data attached? We can see the fitness by the heatmap, but only a couple of residues are selected to check the effects.

- Line 202, Deep mutational scanning data substantiates that the bFP interacts with the host membrane. Could this be quantified?

- Calcium ion could play an important role in the SAR-CoV-2 membrane fusion, and based on Deep mutational analysis, Arg might be important since it does not tolerate any nonnegative charged mutations. Can you elaborate on this in terms of structural aspects or cell tropism?

3. F823 mutation and binding to antibodies

- F823 is a highly conserved amino acid residue and is proposed to be an important LLF motif sequence that embeds into the lipid bilayer. If you make mutations in the LLF motif, there is reduced membrane fusion in in vitro studies. Please elaborate on this.

Reviewer #3 (Remarks to the Author):

In this manuscript, the authors used deep-mutational scanning (DMS) method to study the tolerance of mutations spanning a 47 amino-acid region, 808aa – 855aa, within the SARS-CoV-2 Spike S2 region. This region spans S2' cleavage site utilised by TMPRSS2 protease and a bona fide fusion peptide. Mutated viral libraries (in 2 biological replicates) were generated from BAC DNA-based libraries in Vero cells (passage 0) and then and passaged once in two cells lines, Vero and Calu-3 (passage 1), and then sequenced using NGS. Fitness scores were calculated for each individual mutation in each codon in both cell lines based on comparing data for passage 1 viruses with the data for input BAC DNA-based libraries. Investigators then studied the effect of two mutations at position 813 (S813V and S813K) that had high fitness scores in Vero cells (no or very low TMPRSS2 expression) on utilization of TMPRSS2 protease for S2' cleavage using VSV-Spike pseudoviruses. The mutations were then also introduced into mouse-adapted SARS-CoV-2 and further analysed in Vero cells lines expressing TMPRSS2 and/or ACE2 to analyse the dependency on TMPRSS2 for entry. The same mouse adapted SARS-CoV-2 mutant viruses showed lower virulence in mice causing less weight loss and higher survival rates. The results above demonstrate the important role of S813 in TMPRSS2-mediated virus entry and pathogenesis.

While several mutations in position 813 had high fitness advantage, some mutations at other positions were also well tolerated in Vero cells (e.g, I834M/L, Y837M/H, G838F, etc.) but none of these mutations were investigated in this study.

Antibody neutralization escape mutations were also investigated using two cross-reactive neutralizing antibodies targeting SARS-CoV-2 Spike fusion peptide. A single mutation F823Y conferred antibody resistance to both antibodies and was validated using VSV-Spike pseudoviruses harbouring F823Y mutation.

The use of DMS in live SARS-CoV-2 is certainly novel and the region in Spike protein targeted for DMS (TMPRSS2 cleavage site and fusion peptide) is an interesting choice as it is essential for virus entry. The low mutational tolerability of this region perhaps is not surprising given its essential role in TMPRSS2 cleavage and virus entry. The selection of amino acid 813 that showed high mutation tolerance for analysing mutations is understandable but rather limited in scope given that some mutations in other amino acid position within DMS region were also tolerated, at least in Vero cells. Some other aspects of the work mainly concerning generation and characterization of viral DMS libraries (see comments below) require further experimentation and/or clarification.

Comments:

1. While the raw read counts data are presented in a table online, the diversity and mutation frequencies of input BAC and recovered/passaged virus libraries for each codon should also be shown as codon variant heatmaps and graphs, respectively, perhaps in a supplementary figure. This will allow to better visualise diversities of the input and recovered virus libraries and provide more comprehensive characterization of the libraries.

2. What are the average numbers of mutations per clone/virus? Given the small size of the DMS fragment and the read length chosen, the authors should be able to identify if there are multiple mutations in individual clones/viruses. From most published DMS studies there may be between 1 to 12 mutations per clone/virus; this has not been accounted for in this study which can influence interpretation of the results. In particular, of the mutations that were selected for downstream analysis, did the authors verify or validate that the selected clones/viruses had these single mutations only?

3. There are no data on diversity and mutation frequencies of recovered P0 virus and these data should be used to calculate fitness for P1 viruses rather than doing these calculations based on the data from input BAC libraries. This is important for addressing the likely bottleneck effects caused by potentially inefficient transfection and virus replication/recovery. It is also important to know the P0 virus library diversity to ensure that the low MOI (0.01) used for further passage and under antibody treatments is sufficient to contain all mutants present in the library. The enrichment of mutations then should be calculated by comparing P1 data with P0 data.

4. The formula for calculating relative resistance (Fig 5A and B) should be shown in the main Methods for the analysis of sequencing data as it is ambiguous how these calculations were performed. It would also be informative to show codon variant heatmaps and mutation frequency graphs, as per my comment 1 above.

Reviewer #3 (Remarks on code availability):

The code is well annotated on github and raw data and output tables have been provided. Why we did not reproduce the analysis pipeline using provided code, there is no reason to expect it will not be reproducible.

Reviewer #1 (Remarks to the Author):

Lei, Qing et al. report deep mutational scanning studies on the SARS-CoV-2 spike protein focused on the fusion peptide and adjacent region. In replicating Wuhan-Hu-1 virus generated from reverse genetics, they constructed a mutant library encompassing residues 808-855 which includes the bona fide fusion peptide (816-834), the S2' cleavage site (815/816), and fusion peptide proximal region (but not a putative internal fusion peptide suggested from a recent structure, 867-909). Mutational constraint confirms the importance of the bona fide fusion peptide. They identify mutations with protease-specific effects on viral replication fitness and mutations escaping broadly cross-reactive FP mAbs. This study is to my knowledge the first deep mutational scanning experiment in a fully replicating SARS-CoV-2 virion as opposed to pseudovirus or cell-display platforms, which is a big innovation. The conclusions are interesting and validated nicely. I find this to be a nice and well-performed study.

Response: Thank you for the supportive comment.

Minor comments, in no particular order:

1. Have mutations at position 813 occurred at all during the pandemic, either in named variants or in other sequences seen during viral surveillance (e.g. chronic infection lineages). What about other mutations/sites focused on during this scan?

Response: We thank the reviewer for the comment. We have analyzed 15 million SARS-CoV-2 sequences available on GISAID. Mutations at Spike residue 813 can be observed and are shown in (Figure S6B). In the revised manuscript, we have also noted that:

Lines 123-124: "Of note, while mutations in circulating SARS-CoV-2 have been observed at each of residues 808 to 855, their natural occurrence frequency are all less than 0.3%⁴⁰."

2. Figure 1: "" could be used instead of "-" to denote nonsense mutations ("- would typically represent a gap character i.e. deletion)*

Response: Figure 1, Figure 5, and Figure S4 are modified as suggested.

3. Description of biosafety considerations and cost/benefit analysis of creating replicating mutant SARS-CoV-2 should be considered.

Response: We agree with the reviewer that it is important to include a description of the biosafety considerations, which are now included in the revised manuscript:

Lines 262-267: "The bFP and its flanking regions were chosen for our study here as their role in mediating the fusion process has not been thoroughly characterized. Since no antivirals are currently approved by the Food and Drug Administration to target this region, mutations within this region should not inhibit the efficacy of any clinical regimen available for treating SARS-CoV-2 infection. Furthermore, mutagenesis described in this study was performed in the S2 region of S instead of the immunodominant RBD. As a result, most people should have robust immunity against all the SARS-CoV-2 mutants in our library due to vaccinations or prior infections."

Reviewer #2 (Remarks to the Author):

Lei and colleagues significantly advance our understanding of the effects of mutations in the SARS-CoV-2 spike protein in this paper. They employ deep mutational scanning methods to characterize and pinpoint crucial residues within the SARS-CoV-2 spike protein, utilizing in vivo assays that can be categorized into two distinct sections. One noteworthy finding in the manuscript highlights the relationship between cell tropism and TMPRSS2 cleavage events, emphasizing the role of the S813 residue. Conversely, the F823 mutation is identified as impeding the binding of antibodies. While the manuscript is well-written and represents substantial work, some pertinent issues need to be addressed at this stage. This mainly clarifies and elaborates on novel findings about S813 and the future direction of the discussion.

Response: Thank you for the encouraging and constructive comment.

1. Importance of S813 residue and entry in different cell types

Based on the hypothesis, the Vero cell prefers entering through the endosome, while Calu-3 prefers entering through the plasma membrane because the Vero cell lacks TMPRSS2. Testing the hypothesis of TMPRSS2 overexpression and membrane fusion pathway is very confusing since there is so much data to digest (paragraph lines 134-150). Consider polishing the section to help readers to understand better.

Response: We thank the reviewer for this comment. We have modified the manuscript as suggested to improve clarity (lines 136-157).

- The data show many Vero data assays but only one with Calu-3 in the supplement. It would be beneficial to show Calu-3 data in detail to understand better. I wonder if we should see a reduction in mutants compared to WT since S813 mutants have reduced entry in the presence of TMPRSS2. Figure S2E only shows a relative % entry. Also, what happens when you add FBS to it? Will it reduce entry, as shown in the Vero cell line?

Response: We have included a figure showing that pseudovirus entry levels of Spike with S813V or S813K mutations were reduced in Calu-3 cells when normalized to that of pseudovirus with WT Spike in the revised manuscript (Figure S3F). The result is described in the results section of the revised manuscript:

Lines 155-156: "Consistently, we observed reduction in entry for S813V and S813K in Calu-3 cells, where TMPRSS2 is expressed on the cell surface compared to that of WT (Figure S3F)."

We have also performed pseudovirus entry assay in Calu-3 cells in the presence of FBS. However, we did not observe inhibition of entry in the presence of Calu-3 (Figure R1). This indicates that Calu-3 may have a higher tolerance for TMPRSS2 inhibition than Vero-TMPRSS2 cells. This is further supported by the IC₅₀ values observed in Figure S2C and E. The IC₅₀ values of camostat inhibition for WT, S813V and S813K are 0.16, 0.11 and 0.36 μ M respectively in Vero-TMPRSS2 cell while that in Calu-3 cells are 2.82, 11.68 and 7.50 μ M for WT, S813V and S813K respectively. It is consistent with our hypothesis that the WT and mutant Spikes displayed similar susceptibility to camostat inhibition (Figure 2E) due to the insufficient inhibition of TMPRSS2 in Calu-3 cells as opposed to that in Vero-TMPRSS2 cells, where TMPRSS2 was more effectively inhibited by FBS (Figure S2C and D). The mechanism behind the differences in tolerance for TMPRSS2 inhibition between the two cell lines is unclear. Thus, we prefer not to include the Calu-3 + FBS data (Figure R1) in our manuscript.

Figure R1. Pseudovirus entry in Calu-3 cells without (blue) or with 10% FBS (orange) is shown. Mock: no pseudoparticles, No S: Spike-less pseudoparticles, Wuhan: pseudoparticles coated with SARS-CoV-2 (Wuhan-Hu-1) S.

- In the entry assay, Were VSVpps added based on the protein concentration measured in the western blot? How do you quantify the amount of VSVpps? It should be mentioned since the expression levels of the WT and mutants are different.

Response: The method for quantifying the amount of VSVpps is described in the revised manuscript:

Line 608: "VSVpps were quantified based on VSV-M expression by Western Blot analysis."

- In Figure 2, the entry assay in Vero-TMPRSS2 cells(B) shows that S813V and S813K both reduce but still have similar readings to Vero cells(A). Does this suggest that VSVpps will take both pathways when TMPRSS2 is expressed? It is unclear whether the plasma membrane fusion pathway is preferred or if they still have both pathways take one or another. Please clarify this.

Response: Clarification is made in the revised manuscript:

Lines 146-147: "... , indicating that they all preferred TMPRSS2-mediated entry when TMPRSS2 was overexpressed."

Lines 151-153: "This observation can be explained by the efficient endosomal entry of S813V and S813K when TMPRSS2-mediated entry is highly suppressed."

- Please explain and discuss all of the data presented such as the discussion about Figure S2D will be helpful. I wonder if it is due to less sensitivity or more endosomal fusion.

Response: As also stated in the response above, we have added a sentence to clarify our interpretation of the observation in Figure S3D (previously Figure S2D):

Lines 151-153: “This observation can be explained by the efficient endosomal entry of S813V and S813K when TMPRSS2-mediated entry is highly suppressed.”

- If it is possible, it might be interesting to see if you can correlate Figure 3 data with Figure 2.

Response: We thank the reviewer for this comment. However, we are not sure if we have interpreted this comment correctly. We interpreted this comment in two ways:

1. Correlating the data in Figure 2 and 3. This will indicate how well the pseudovirus system agrees with the authentic SARS-CoV-2 infection system. Based on our data, we believe that these two systems agree with each other, but we prefer not to include this as comparison between the two systems is not the focus of this study.
2. Correlating the Vero vs Vero-TMPRSS2 data independently for Figure 2 and 3. We showed the data in Figure 2 and 3 to compare the difference between WT or mutant entry in these cells to show the inability of the mutants to efficiently utilize TMPRSS2. Correlating entry for Vero and Vero-TMPRSS2 would be an alternative way for presenting such data. Since we believe our revised manuscript already highlights this difference, we prefer not to include a correlation plot. However, we will include a correlation plot if requested by the reviewer or editor.

2. Low mutational tolerance of the bFP

- Line 185, showed differential fitness effects between Calu-3 and Vero cells. I am not clear other than the heat map. Is there any other effect data attached? We can see the fitness by the heatmap, but only a couple of residues are selected to check the effects.

Response: We thank the reviewer for this comment. A major goal of this manuscript is to establish a deep mutational scanning system for characterizing the phenotypes of multiple SARS-CoV-2 mutants in parallel. Our screening data were obtained from two independently generated libraries which correlated with each other (Figure S2), indicating the high degree of reproducibility of the system. The reason we focused on residue 813 for experimental validation is due to the stark contrast in replication fitness of the mutants observed in Calu-3 and Vero cells and that S813 is directly upstream of the S2' cleavage site (R815). This led to our hypothesis that residue 813 may affect TMPRSS2 sensitivity due to its proximity to the cleavage site which we further investigated. In the discussion section of the revised manuscript, we have also mentioned the implications of other mutations that show differential fitness effects between Calu-3 and Vero cells:

Lines 302-305: “At the same time, mutations with differential fitness effects between Calu-3 and Vero cells that are distal from the S2' cleavage site may represent mechanisms unrelated to S2' cleavage. We believe those mutations, albeit outside of the scope of this study, warrant future investigations.”

- Line 202, Deep mutational scanning data substantiates that the bFP interacts with the host membrane. Could this be quantified?

Response: NMR is the most direct approach to measure the interaction between the bFP and the membrane (ref #13 in our manuscript). However, it is technically very challenging to setup such NMR experiment. Other potentially feasible approaches are biolayer interferometry (BLI) and surface plasmon

resonance (SPR). However, their feasibility is yet to be shown. In fact, there is a lack of simple approaches to quantify protein-lipid interaction in general.

- Calcium ion could play an important role in the SAR-CoV-2 membrane fusion, and based on Deep mutational analysis, Arg might be important since it does not tolerate any nonnegative charged mutations. Can you elaborate on this in terms of structural aspects or cell tropism?

Response: There are two Arg residues in our region of interests, namely R815 and R847. R815 can only tolerate mutation to Lys, which is a positively charged amino acid. On the other hand, R847 can tolerate mutations to Ser and Thr, which are small polar amino acids. Therefore, our deep mutational analysis actually did not identify any Arg residue that cannot tolerate non-negative charged mutations (Figure 1).

3. F823 mutation and binding to antibodies

- F823 is a highly conserved amino acid residue and is proposed to be an important LLF motif sequence that embeds into the lipid bilayer. If you make mutations in the LLF motif, there is reduced membrane fusion in in vitro studies. Please elaborate on this.

Response: Based on our deep mutational scanning experiment, F823Y has a neutral effect on replication fitness (Figure 1). As a result, while F823Y may reduce membrane fusion in in vitro studies, it does not seem to affect replication fitness. Consistently, F823Y naturally exists in bat betacoronavirus HKU9, as stated in the Discussion section:

Line 312-314: “F823Y is a natural variant in bat betacoronavirus HKU9 and is also observed in circulating SARS-CoV-2 at a very low frequency (Figure S7).”

Reviewer #3 (Remarks to the Author):

In this manuscript, the authors used deep-mutational scanning (DMS) method to study the tolerance of mutations spanning a 47 amino-acid region, 808aa – 855aa, within the SARS-CoV-2 Spike S2 region. This region spans S2' cleavage site utilised by TMPRSS2 protease and a bona fide fusion peptide. Mutated viral libraries (in 2 biological replicates) were generated from BAC DNA-based libraries in Vero cells (passage 0) and then and passaged once in two cells lines, Vero and Calu-3 (passage 1), and then sequenced using NGS. Fitness scores were calculated for each individual mutation in each codon in both cell lines based on comparing data for passage 1 viruses with the data for input BAC DNA-based libraries. Investigators then studied the effect of two mutations at position 813 (S813V and S813K) that had high fitness scores in Vero cells (no or very low TMPRSS2 expression) on utilization of TMPRSS2 protease for S2' cleavage using VSV-Spike pseudoviruses. The mutations were then also introduced into mouse-adapted SARS-CoV-2 and further analysed in Vero cells lines expressing TMPRSS2 and/or ACE2 to analyse the dependency on TMPRSS2 for entry. The same mouse adapted SARS-CoV-2 mutant viruses showed lower virulence in mice causing less weight loss and higher survival rates. The results above demonstrate the important role of S813 in TMPRSS2-mediated virus entry and pathogenesis.

While several mutations in position 813 had high fitness advantage, some mutations at other positions were also well tolerated in Vero cells (e.g, I834M/L, Y837M/H, G838F, etc.) but none of these mutations were investigated in this study.

Antibody neutralization escape mutations were also investigated using two cross-reactive neutralizing

antibodies targeting SARS-CoV-2 Spike fusion peptide. A single mutation F823Y conferred antibody resistance to both antibodies and was validated using VSV-Spike pseudoviruses harbouring F823Y mutation.

The use of DMS in live SARS-CoV-2 is certainly novel and the region in Spike protein targeted for DMS (TMPRSS2 cleavage site and fusion peptide) is an interesting choice as it is essential for virus entry. The low mutational tolerability of this region perhaps is not surprising given its essential role in TMRSS2 cleavage and virus entry. The selection of amino acid 813 that showed high mutation tolerance for analysing mutations is understandable but rather limited in scope given that some mutations in other amino acid position within DMS region were also tolerated, at least in Vero cells. Some other aspects of the work mainly concerning generation and characterization of viral DMS libraries (see comments below) require further experimentation and/or clarification.

Response: Thank you for the constructive comments, which help us improve the manuscript. We have revised our manuscript accordingly (see responses below).

Comments:

1. While the raw read counts data are presented in a table online, the diversity and mutation frequencies of input BAC and recovered/passaged virus libraries for each codon should also be shown as codon variant heatmaps and graphs, respectively, perhaps in a supplementary figure. This will allow to better visualise diversities of the input and recovered virus libraries and provide more comprehensive characterization of the libraries.

Response: In the revised manuscript, we have included heatmaps that show the frequencies of individual codon variants in different samples, including the input BAC mutant library as well as passaged virus libraries (Figure S1).

2. What are the average numbers of mutations per clone/virus? Given the small size of the DMS fragment and the read length chosen, the authors should be able to identify if there are multiple mutations in individual clones/viruses. From most published DMS studies there may be between 1 to 12 mutations per clone/virus; this has not been accounted for in this study which can influence interpretation of the results. In particular, of the mutations that were selected for downstream analysis, did the authors verify or validate that the selected clones/viruses had these single mutations only?

Response: Unlike most published DMS studies, our saturation mutagenesis library was constructed in a way that no secondary mutations would be introduced. A schematic of our PCR-based mutagenesis method, from Figure S1 in our previous study (PMID: 36417523), is shown below. Each cassette of primers was used to perform mutagenesis separately. Cassette primers were not mixed to prevent secondary mutations. In other words, mutations were introduced at eight consecutive positions (one cassette) per each PCR, and the PCR products from different cassettes were mixed at the last step after the overlap PCRs.

Figure R2. Schematic diagram from Figure S1 of Ouyang et. al, Sci Adv, 8, eadd7221 (2022). A total of 36 forward PCRs and 36 reverse PCRs were performed. Subsequently, 36 overlap PCRs were performed to ensure each molecule in the final PCR product contained only 1 amino-acid mutation.

3. There are no data on diversity and mutation frequencies of recovered P0 virus and these data should be used to calculate fitness for P1 viruses rather than doing these calculations based on the data from input BAC libraries. This is important for addressing the likely bottleneck effects caused by potentially inefficient transfection and virus replication/recovery. It is also important to know the P0 virus library diversity to ensure that the low MOI (0.01) used for further passage and under antibody treatments is sufficient to contain all mutants present in the library. The enrichment of mutations then should be calculated by comparing P1 data with P0 data.

Response: When we were setting up this deep mutational scanning platform, we did a pilot experiment that included sequencing of P0 virus mutant library. Our data indicated that selection has already started in P0, during virus rescue, as evidenced by the low P0-derived fitness values (i.e. comparing P0 to input) of nonsense mutations (Figure R3). This result is not surprising because our virus rescue was performed by transfecting Vero cells, which not only produced recombinant SARS-CoV-2 viruses from the transfected BAC library, but are also susceptible to infection by nascent SARS-CoV-2 viruses in the supernatant. Therefore, our fitness values in the present study were derived from comparing P1 with the input BAC library. In fact, it is a common practice to compute fitness values by comparing P1 or P2 to the input plasmid library in deep mutational scanning studies using recombinant virus systems, including influenza virus (PMID: 30104379; PMID: 36640354), hepatitis C virus (PMID: 24722365), human immunodeficiency virus (PMID: 27959955), and Zika virus (PMID: 30886357; PMID: 31511387).

Figure R3. We computed the fitness values for individual mutations using the P0 data in the same way as we did in the manuscript for P1. **(A)** The fitness values are shown as a heatmap. **(B)** The distributions of P0-derived fitness values for missense mutations (mis), nonsense mutations (non), and silent mutations (sil) are shown as a stripchart overlaid with a violin plot (right panel). **(C)** Correlation of fitness values for individual mutations between two biological replicates of P0 is shown as a scatterplot.

4. The formula for calculating relative resistance (Fig 5A and B) should be shown in the main Methods for the analysis of sequencing data as it is ambiguous how these calculations were performed. It would also be informative to show codon variant heatmaps and mutation frequency graphs, as per my comment 1 above.

Response: The formula for calculating relative resistance was mistakenly omitted from our first submission, but is now shown in the Methods section of the revised manuscript (lines 558-560):

Relative resistance (R) for a given mutant i against antibody a in cell line c was computed as follows:

$$R_{i,c,a} = W_{i,c,a} - W_{i,c,no\ antibody}$$

Moreover, the codon variant heatmaps for all the passaged virus libraries, both with and without antibody selections, are shown in Figure S1.

REVIEWERS' COMMENTS

Reviewer #2 (Remarks to the Author):

I appreciate that the authors addressed all questions and queries, and I do not have further comments. The modified paper is acceptable for publication.

Reviewer #3 (Remarks to the Author):

Point 1. Happy with the response and inclusion of heatmaps as supplementary Fig 1.

Point 2. I appreciate that the saturation mutagenesis methods used by authors in this manuscript and in the previous Science Advances paper could minimise secondary mutations for the DMS library construction. I also appreciate the figure in the rebuttal to clarify this. However, it is untrue that secondary mutations are not introduced using this method, and it has not been computed empirically both here and in the Science Advances manuscript.

To demonstrate this, I have taken the liberty of examining the first 10 reads in the input DNA sequencing library (by no means a rigorous sample) and the Vero E6 no antibody replicate 1 (see attached Figure) and identified a range of 0-5 non-synonymous mutations (average 1.5) for the input DNA library and between 0-4 non-synonymous mutations for the Vero library (average 2.3). I'd argue that these mutations per clone or read are within reasonable or tolerable expectations for DMS studies. Still, it is untrue to claim that the methods used here and in the previous Science Advances paper only generate one mutation per clone and I believe quality metrics such as number of mutations per read or clone (using PEARED reads) would be valuable to establish mutation ranges for DMS studies and should be calculated for all your libraries.

Point 3. As I suspected and as your pilot data in the rebuttal show, selection already occurred in P0- derived virus. This could be due to selective propagation of beneficial mutants but also due to transfection bottleneck. While it is true that some viral DMS studies used P1 or P2 viruses to identify enriched mutations (or fitness values), other studies used P0 viruses and suggested that bottleneck could be contributing to occurring selection (e.g. PMID: 30886357). I strongly suggest including pilot data at least as a supplementary figure and clearly articulate the likely bottleneck effect as a potential limitation for recovery of highly diverse DMS viral libraries. I would also still argue that fitness values following selection at least under antibody treatment would be more accurately represented if they are calculated in passaged viruses relative to P0 virus data and not to Bacmid data. Simply because the starting material used for antibody selection (P0 virus) already had greatly reduced diversity of mutant virus library.

Reviewer #3 (Remarks on code availability):

as per my initial review

Reviewer #3 (Remarks to the Author):

Point 1. Happy with the response and inclusion of heatmaps as supplementary Fig 1.

Response: Thank you for this supportive comment.

Point 2. I appreciate that the saturation mutagenesis methods used by authors in this manuscript and in the previous Science Advances paper could minimise secondary mutations for the DMS library construction. I also appreciate the figure in the rebuttal to clarify this. However, it is untrue that secondary mutations are not introduced using this method, and it has not been computed empirically both here and in the Science Advances manuscript. To demonstrate this, I have taken the liberty of examining the first 10 reads in the input DNA sequencing library (by no means a rigorous sample) and the Vero E6 no antibody replicate 1 (see attached Figure) and identified a range of 0-5 non-synonymous mutations (average 1.5) for the input DNA library and between 0-4 non-synonymous mutations for the Vero library (average 2.3). I'd argue that these mutations per clone or read are within reasonable or tolerable expectations for DMS studies. Still, it is untrue to claim that the methods used here and in the previous Science Advances paper only generate one mutation per clone and I believe quality metrics such as number of mutations per read or clone (using PEAREd reads) would be valuable to establish mutation ranges for DMS studies and should be calculated for all your libraries.

Response: The analysis performed by the reviewer focused on the first 10 reverse reads and did not account for the forward reads. If we look at the sequencing quality of the reverse read of the BAC mutant library (i.e. input DNA library, SRR22668019), we can see that the quality in the first ~20,000 reverse reads of the BAC mutant library is relatively low (**Figure R1**). The mean Phred quality scores of first 10 reverse reads are 28, 33, 26, 33, 34, 32, 31, 27, 32, and 33, all of which are below the median (median Phred quality score of reverse reads of the BAC mutant library = 35). As a result, it is not surprising that their sequences do not align well with their corresponding forward reads (**Figure R2**). These observations indicate that the first 10 reverse reads are not an accurate representation of our mutant library due to their low sequence quality.

Figure R1. Each data point represents the mean Phred quality score for each of the first 50,000 reverse reads of the BAC mutant library (i.e. input DNA library). Red line represents a smooth curve based on moving average.

However, we agree that analyzing the number of mutations per read is valuable. This analysis is included as Figure S3C and described in results section of the revised manuscript:

```
Query 1 TTTGGTGGTTTTAATTTTTTCAAAATATTACCAGATCCATCAAAACCAAGCAAGAGGTC 60
Sbjct 228 TTTGGTGGATTAAATTTTTTCAAAATATTACCAGATCCATCAAAACCAAGCAAGAGGTC 169
Query 61 TTTATTGAAGATCTACTTTTCAACAAAGTGACACTTGACAGATGCTGGCTTCATCAACAAA 120
Sbjct 168 TTTATTGAAGATCTACTTTTCAACAAAGTGACACTTGACAGATGCTGGCTGCATCAACAAA 109
Query 121 TATGGTGATTGCGCTCGGTTTTATCGCTGCTAGAGACCTCATTGTGTCACAAAAGTTTAA 180
Sbjct 108 TATGGTGATTGCGCTAGGTGATATCGCTGCTAGAGACCTCATTGTGTCACAAAAGTTTAA 49
Query 181 GGCTTACTGTTTTGCCACCTTTTCTCACAGATGAAATGATTGCTCAA 229
Sbjct 48 GGCTTACTGTTTTGCCAACTTTGCTCACAGATGAAATGATTGCT-AAA 1
```

Lines 124-126: "Of note, our mutant library construction

Figure R2. Sequence alignment between the forward read (SRR22668019.1.1) and reverse read (SRR22668019.1.2) of the first read in the input DNA library.

approach was designed to prevent secondary mutations (see Methods). Consistently,

>90% of the sequencing reads of the BAC mutant library had either 0 or 1 mutation (Figure S3C).”

To provide additional validation, we sequenced 20 clones of the BAC mutant library. This result is included as Figure S3D and described in the results section of the revised manuscript:

Lines 126-130: “To assess the mutation rate of the BAC mutant library without confounding by sequencing errors from next-generation sequencing, we sequenced 22 individual clones of the BAC mutant library. Among the 22 clones, 20 (91%) had 0 or 1 mutation, two clones (9%) had 2 mutations, and none had >2 mutations (Figure S3D). This result substantiates that our mutagenesis approach yielded predominantly 1 mutation per clone.”

Point 3. As I suspected and as your pilot data in the rebuttal show, selection already occurred in P0- derived virus. This could be due to selective propagation of beneficial mutants but also due to transfection bottleneck. While it is true that some viral DMS studies used P1 or P2 viruses to identify enriched mutations (or fitness values), other studies used P0 viruses and suggested that bottleneck could be contributing to occurring selection (e.g. PMID: 30886357). I strongly suggest including pilot data at least as a supplementary figure and clearly articulate the likely bottleneck effect as a potential limitation for recovery of highly diverse DMS viral libraries. I would also still argue that fitness values following selection at least under antibody treatment would be more accurately represented if they are calculated in passaged viruses relative to P0 virus data and not to Bacmid data. Simply because the starting material used for antibody selection (P0 virus) already had greatly reduced diversity of mutant virus library.

Response: There is a clear difference in low P0-derived fitness distributions between nonsense mutations and silent mutations ($P < 0.0001$). This difference can only be explained by fitness selection but not transfection bottleneck. If transfection bottleneck happens without fitness selection, P0-derived fitness distributions of nonsense mutations and silent mutations are expected to be the same. However, we do agree that transfection bottleneck exists, as indicated by the imperfect correlation P0-derived fitness values between replicates. In the revised manuscript, we have included the data of P0 as Figure S3A-B and described the genetic bottleneck at the transfection step:

Lines 117-124: “To further characterize our BAC-based reverse genetic system for deep mutational scanning, we also analyzed the post-transfection mutant library by next-generation sequencing. The result indicated that fitness selection was present at the transfection step as the fitness values of silent mutations were significantly higher than nonsense mutations in the post-transfection mutant library ($p < 0.0001$, Figure S3A). Besides, the fitness values of the post-transfection mutant library had a Pearson correlation coefficient of 0.68 between replicates (Figure S3B), suggesting that a genetic bottleneck took place at the step of transfection, albeit mild, in our deep mutational scanning experiments.”

The calculation of relative resistance involves comparison of fitness under antibody selection versus fitness without antibody selection. For reference, here is the formula for relative resistance:

$$\text{Relative resistance} = \text{normalized} \left(\log_{10} \frac{\text{frequency}_{\text{post-selection, with antibody}}}{\text{frequency}_{\text{Bac mutant library}}} \right) - \text{normalized} \left(\log_{10} \frac{\text{frequency}_{\text{post-selection, no antibody}}}{\text{frequency}_{\text{Bac mutant library}}} \right)$$

The first term $normalized\left(\log_{10}\frac{frequency_{post-selection, with\ antibody}}{frequency_{Bac\ mutant\ library}}\right)$ represents $fitness_{with\ antibody}$ and the second term $normalized\left(\log_{10}\frac{frequency_{post-selection, no\ antibody}}{frequency_{Bac\ mutant\ library}}\right)$ represents $fitness_{no\ antibody}$. Treating BAC mutant library as the input (i.e. demoninator) for calculating $fitness_{no\ antibody}$ and P0 virus as the input for $fitness_{with\ antibody}$ will lead to two different denominators in the first and second terms, which in turn will make it difficult to interpret the biological meaning of relative resistance. Therefore, we believe it is more appropriate to use BAC mutant library for the calculation of $fitness_{with\ antibody}$.